# Improvement Prediction on the Dynamic Performance of Epoxy Composite Used in Packaging by Using Nano-Particle Reinforcements in Addition to 2-Hydroxyethyl Methacrylate Toughener

**DOI:** 10.3390/ma14154193

**Published:** 2021-07-27

**Authors:** Chih-Ming Chen, Huey-Ling Chang, Chun-Ying Lee

**Affiliations:** 1Department of Mechanical Engineering, National Chin-Yi University of Technology, Taichung 41170, Taiwan; cmchentc@gmail.com; 2Department of Chemical and Materials Engineering, National Chin-Yi University of Technology, Taichung 41170, Taiwan; 3Graduate Institute of Manufacturing Technology, National Taipei University of Technology, Taipei 10608, Taiwan; leech@mail.ntut.edu.tw

**Keywords:** electronic packaging, dynamic storage modulus, loss tangent, optimization

## Abstract

Epoxy with low viscosity and good fluidity before curing has been widely applied in the packaging of electronic and electrical devices. Nevertheless, its low flexibility and toughness renders the requirement of property improvement before it can be widely acceptable in dynamic loading applications. This study investigates the possible use of 2-hydroxyethyl methacrylate (HEMA) toughening agent and nano-powders, such as alumina, silicon dioxide, and carbon black, to form epoxy composites for dynamic property improvement. Considering the different combinations of the nano-powders and HEMA toughener, the Taguchi method with an L9 orthogonal array was adopted for composition optimization. The dynamic storage modulus and loss tangent of the prepared specimen were measured by employing a dynamic mechanical analyzer. With polynomial regression, the curve-fitted relationships of the glass transition temperature and storage modulus with respect to the design factors were obtained. It was found that although the raise in the weight fraction of nano-powders was beneficial in increasing the rigidity of the epoxy composite, an optimal amount of HEMA toughener existed for its best damping improvement.

## 1. Introduction

Epoxy is one of widely used synthesized resins for general purposes and industrial applications. Due to its low viscosity and good fluidity before curing and excellent strength and stiffness after curing, epoxy resin is commonly thought of as a good candidate for packaging use in electrical and electronic applications. However, with the increase in degree of crosslinking among its molecular chains after the reaction with hardener agent finished, the cured epoxy often becomes too brittle to be applied in some circumstances which require material toughness, such as the environments involving dynamic loading or vibrations.

By adding some “reinforcements” into the epoxy resin to form composites, the mechanical properties of the resin can be tuned. Liu and coworkers [1] investigated the effects on fracture toughness by adding nano-powders of silicon dioxide and rubber into the epoxy resin. The uni-axial tensile performance of the composite revealed both the increase in Young’s modulus and fracture toughness. The incorporation of nano-rubber powder demonstrated a clear improvement on its fracture toughness. The other SiO_2_ nano-powder has 3D network molecular structure and forms a floccular ball-like particle, which is commonly used as a reinforcement in composite [2]. The combination of SiO_2_ nano-powder and glass fiber in a composite laminate has been reported to raise the fatigue life by 3–4 folds compared to its pristine epoxy resin counterpart [3]. The study by Chen et al. [4] also reported the increase in both stiffness and fracture toughness of epoxy composite by adding spherical SiO_2_ nano-powders. However, the glass transition temperature was lowered if the content of SiO_2_ nano-powder was greater than 5 wt.%.

In addition to simply adding inorganic powder for reinforcement, the interface usually needs treatment to improve the compatibility between phases. Liu and coworkers [5] used MXene (Ti_3_AlC_2_ and Ti_3_C_2_Tx) to treat the acidified short graphite fibers and reported the 100% and 67% increases in tensile and flexural strengths of the epoxy composite, respectively. The SiO_2_ nano-powders also had good compatibility with the epoxy resin and its epoxy composite showed the improvements in tensile compressive strengths and fracture toughness [6,7]. Moreover, these well dispersed nano-powders between the reinforced fibers and the resin even presumed as the buffers for absorbing energy in the interfacial bonding [8].

The other approach for toughening the epoxy resin is by modifying its structure with interpenetrating polymer networks (IPNs). This structural modification down to the molecular level with uniform interlacing of toughening phase and the crosslinked epoxy phase could promote the material compatibility. Around the 1980s, Sperling and coworkers [9,10] used different monomers and polymers to prepare toughened epoxy with different types of IPNs, such as full-IPNs and semi-IPNs. They also studied the chemical compositions, microstructural morphology, phase behavior and the correlation with their mechanical property by employing various precision instruments. It was reported that the toughener and the epoxy resin were uniformly mixed at the molecular level or in microscopic phase separation. Since there was no chemical bonding between these two phases, the individual properties from these constituents were to be retained. However, since these two phases were intertwined closely, the cooperative effect emerged as an improvement in its toughness over its mechanically mixed counterpart.

The further modification on the microstructure of IPNs was proposed by Hsieh and Han [11] with the introduction of PU based on polybutylene adipate, PU(PBA) and PU based on polyoxypropylene, PU(PPG), into diglycidyl ether bisphenol A, DGEBA, to form grafted IPNs. The tensile test result revealed that the grafted IPNs could increase the α-and β-transition domain of the epoxy and PU and, subsequently, its tensile strength. Lin and Lee [12] based on the DGEBA matrix added ethylene dimethacrylate and 2-hydroxymethyl methacrylate (HEMA) to form full-IPNs and semi-IPNs. Their experimental results showed the material with full-IPNs had a higher Young’s modulus, larger percent elongation and better fracture toughness to absorb more energy in dynamic loading. When the added reinforcement was the nano-powder, the epoxy composite not only had increases in tensile strength but also the toughness in impact.

Mimura and coworkers [13] reported the preparation of toughened epoxy with semi-IPNs by using synchronized polymerization of polyethersulfone (PES) and epoxy resin. A 60% increase in fracture toughness was obtained. Sometimes, the addition of toughener could create non-uniform phase separation and crosslinking density. In the work by Kwon and coworkers [14], the appearance of additional side peak in the tanδ of a viscoelastic measurement on the polytriazoleketone (PTK) and polytriazolesulfone (PTS) toughened epoxy resin demonstrated the probable concern.

In the electronic packaging applications, the use of different materials to tune the mechanical property was a common practice. However, the collateral effect arisen from the mismatch in the thermal expansion of its constituents could worsen or even fail the packaging function [15]. The other concern may lie on the viscosity of resin. Oh and coworkers [16] used two micro-fillers, Al_2_O_3_ and SiO_2_, to tune the thermal conductivity of the epoxy resin. However, considering the Al_2_O_3_ filler had higher specific gravity than SiO_2_, less volumetric fraction of the former could have similar weight fraction with the latter. The partial substitution of Al_2_O_3_ for the SiO_2_ would keep the epoxy resin lower in viscosity or higher in fluidity, which was beneficial in packaging. The other report by Khalil et al. [17] illustrated the incorporation of Al_2_O_3_ nano-powder within 2.0 wt.% in epoxy would improve the wetting behavior and tensile shear strength of the epoxy resin. Thus, the addition of nano-powders within 2.0 wt.% in the epoxy resin was adopted in this study.

As reviewed from the previous studies, the incorporation of inorganic nano-powders in epoxy resin seems beneficial to its packaging applications. However, the underlined mechanisms have not been well investigated. Moreover, the reliability performance of epoxy composites with different compositions are susceptible to the service temperature and loading conditions. Therefore, by using the dynamic mechanical analyzer, the main focus of this study is to investigate the viscoelastic behavior and energy dissipation from the internal friction within its microstructure of the epoxy composite over a specified temperature range. Three nano-powders of Al_2_O_3_, SiO_2_, and carbon black were chosen to be the reinforcements of the epoxy resin in addition to a HEMA toughener. The design of experiments by using the Taguchi method was adopted and the analysis of variables (ANOVA) was performed to find the optimal compositions for the dynamic property improvement. The prediction of the dynamic properties of the prepared epoxy composite was established for the design purpose in packaging.

## 2. The Theory

### 2.1. The Reaction Mechanism of Epoxy

In this study, a simultaneous polymerization reaction was adopted to prepare the epoxy with IPNs. Three reactions were proceeded simultaneously in the crosslinking of epoxy, as presented in Figure 1. Firstly, the crosslinking started with the ring-opening reaction of the epoxide group in epoxy and the primary amine of hardener agent to form the hydroxyl group and the formation of secondary amine from the primary amine of the hardener as described in Figure 1a. The thus formed secondary amine could also take part in the ring-opening reaction with the epoxide group as denoted in Figure 1b. In the meantime, the hydroxyl group started a self-catalyzed ring-opening reaction as denoted in Figure 1c, which is also known as etherification, and was the least reactive among the three mentioned reactions. Subsequently, a crosslinkage in spatial dimensions was obtained with the described reactions.

Secondly, benzoyl peroxide as a thermal initiator disintegrated into two free radicals upon being heated and are described in Figure 2. The generation of these free radicals induced the polymerization of 2-hydroxyethyl methacrylate to form the PHEMA polymer as denoted in Figure 3. These PHEMAs constituted the interpenetrating polymer network in the already spatially crosslinked structure of epoxy. Due to the thermoplastic property of the PHEMA, the thus prepared epoxy resin could improve the toughness over the pristine brittle nature.

### 2.2. Dynamic Mechanical Property

A dynamic mechanical analyzer (TA DMA 2980) was used in this study to measure the dynamic properties of the prepared specimens in different time, temperature and loading frequency. In the measurement, the specimen was installed in a specified configuration, such as three-point bending and cantilevered fixture, and was subjected to a vibrational stress or strain. The responses of the specimen in deformation or loading were recorded both in magnitude and phase. Accordingly, the storage modulus E′, loss modulus E″, and coefficient of loss tangent tanδ of the specimen can be calculated as follows.

Storage modulus E′ is calculated as follows.
(1)$$E^{\prime }=\left(\frac{\sigma }{\varepsilon }\right)\cos\delta$$

The loss modulus E″ is calculated as follows.
(2)$$E^{\prime \prime }=\left(\frac{\sigma }{\varepsilon }\right)\sin\delta$$

The coefficient of loss tangent, tanδ, is calculated as follows.
(3)$$\tan\delta =\frac{E^{\prime \prime }}{E^{\prime }}$$

In the above equations, ε and σ represent the amplitudes of the applied strain and the measured stress, respectively. The larger the peak of tan, the more viscous the material behaves. In other words, the specimen with larger tanδ has more damping in response to dynamic loading. Moreover, the temperature where the tanδ attains its peak magnitude denotes the corresponding glass transition temperature of the specimen.

### 2.3. Design of Experiment Using Taguchi Method

Table 1 presents the L_9_(3^4^) orthogonal table for this study. There were four design factors, i.e., the weight contents of A: nano-alumina powder; B: nano-silica powder; C: nano-carbon black powder; and D: HEMA toughener, respectively. Three levels for each factor were selected, which were 0, 1, and 2 wt.% and 0, 5, and 10 wt.% for the nano-powders and the toughener, respectively. Accordingly, the dynamic properties of the specimens were measured by using the DMA analyzer and the S/N ratios were calculated from the quality equation.

The-higher-the-better (HB) characteristics were employed with decibels (db) as its unit, shown in the following.
(4)$$S/N_{\mathrm{LTB}}=-10\times \log_{10}\left(\frac{1}{n}\overset{n}{\underset{i=1}{\sum }}\frac{1}{y_{i}^{2}}\right)$$

### 2.4. Coefficient of Variance

Three specimens for each measurement were employed in this study in order to evaluate the variation of the results. Therefore, the coefficient of variance (C.V.) was calculated accordingly:(5)$$\;C.V.=\frac{S}{\underline{X}}$$
where X and S are the mean and standard deviation for each set of measurements.

### 2.5. Multiple Regression Analysis

With the storage moduli as a substitute for the machining process factor, the regression fit of the object function could be preceded. For simplicity, the dependence of the object function on the factors was assumed to be quadratic and the coupling among the factors was neglected. These assumptions will be checked with the verification example presented later in this study. Thus, the regression function Y is written as follows:(6)$$\;Y=\overset{G}{\underset{i=A}{\sum }}\left(c_{i2}X_{i}^{2}+c_{i1}X_{i}\right)+c_{0}$$
where Xi denotes the value of the ith factor and c_i2_ and c_i1_ are the coefficients for the quadratic and linear terms, respectively. The constant term for the relationship is summed up in c_0_. The object functions that will be evaluated later include the storage moduli.

### 2.6. Coefficient of Determination for Regression

The validity of the regression function needs to be examined. Herein, a coefficient of determination R^2^ was adopted. The larger the R^2^ is, the more accurate the regression function becomes. The coefficient of determination was defined as the ratio of regression variance to total variance. It can be expressed as follows:(7)$$R^{2}=\frac{{\mathrm{SS}}_{T}}{{\mathrm{SS}}_{R}}$$
where SS_T_ is the total sum of squares and SS_R_ is the regression sum of squares. The magnitude of R^2^ lies between 0 and 1, i.e., 0 < R^2^ < 1. The closer the R^2^ approaches 1, the better the regression function represents the physical relationship.

## 3. Experimental

### 3.1. Materials

A diglycidyl ether of bisphenol A type epoxy resin (an epoxide equivalent weight of 180 g/equiv.) was purchased from Chang-Chun Plastics Co. Ltd. (Taipei, Taiwan). The epoxy resin was cured with the incorporation of an amine type epoxy curing agent (the amine hydrogen weight of 65 g/equiv. and amine value 430 g/equiv.). When the nano-powder used was a silicon dioxide (silica) powder, the surface of the fumed silica was chemically modified with a poly(dimethyl siloxane) coupling agent. The hydrophobic fumed silica had a specific surface area (BET) of 100 m^2^/g, an average primary particle diameter of 14 nm, and a tapped density (according to DIN ISO 787/XI, August 1983) of 60 g/L. For the other nano-powders, the aluminum oxide (γ-Al_2_O_3_, alumina) had an average particle diameter of 50 nm and density 2.414 g/cm^3^. The nano carbon black powder was with a specific surface area (BET) of 90 m^2^/g, average particle diameter of 28 nm, and density of 1.719 g/cm^3^. Moreover, the toughening agent was 2-hydroxyethyl methacrylate (HEMA) with a mass density of 1.106 g/mL, which was an acrylate with polar substations that was purchased from TCI Co. Ltd. Benzoyl peroxide (BPO) was used as a thermal initiator. It should be mentioned that all the purchased materials were used as-received without further purification.

### 3.2. Specimen Preparation

Firstly, the epoxy resin and HEMA toughener with specified weight fraction were put in a container of a deaerator and thoroughly mixed. Subsequently, the nano-powders with specified weight fractions were added and agitated mechanically. The well mixed solution, which was mixed with an ultrasonic and centrifugal mixer, was obtained as the pre-mixture for next process. In our previous study [18], the prepared material was found to have a well dispersion of nano-powders in the epoxy matrix. On the other hand, a specified amount of hardener and thermal initiator were mixed in the other container. This mixed hardener was then added into the epoxy pre-mixture and agitated in the centrifugal mixer. Subsequently, the prepared mixture was poured into a DMA specimen mold manufactured according to the ASTM D4065 standard. This casted mold was cured in a 120 °C oven for 1 h and the removed specimen was then post-cured at 140 °C for another 1 h. The prepared specimen has to be placed at room temperature for more than 24 h before it can be employed in the material testing.

Table 2 lists the compositions of the 9 types of specimens adopted from an L_9_(3^4^) orthogonal array of the Taguchi method. Four control factors were employed in the experiments: nano-alumina powder, nano-silica powder, nano-carbon black powder, and HEMA toughener. Three levels for each factor were selected, which were 0, 1, 2 wt.% and 0, 5, 10 wt.% for the nano-powders and the toughener, respectively.

### 3.3. DMA Measurements

Dynamic mechanical analysis (DMA) was performed according to ASTM D4065-01 to determine storage modulus (E′), loss modulus (E′, loss tangent (tanδ), and glass transition temperature (Tg) of the epoxy composites. The tests were conducted in the dual cantilever beam mode with a vibration frequency of 1 Hz and a displacement amplitude of 10 μm in a DMA analyzer (DMA 2980, TA Instruments). The temperature was ramped from 30 to 140 °C at a rate of 2 °C/min. At least three specimens of each type were tested and the data were analyzed. The DMA testing purpose was to simulate the loading conditions of the epoxy composite used in the packaging and subjected to periodic vibration from the surroundings. 

## 4. Results and Discussion

### 4.1. The Dynamic Properties at Different Temperatures

Figure 4 presents the dynamic properties of Specimen T1 measured with the temperature sweeping from 30 °C to 140 °C. There are three results for each specimen: storage modulus, loss modulus, and loss tangent. As mentioned previously, there were three specimens denoted as T1-1, T1-2, and T1-3 that were tested for each specimen type of the design factors specified in Table 1. It is obviously observed that the test results had good repeatability. As observed in the figure, the storage modulus E′ decreased sharply when the temperature reached near 60 °C while the loss modulus E″ increased to a maximum and dropped subsequently. According to these results, the loss tangent defined as tanδ = E″/E′ attained a maximum around 83 °C. The temperature at which the loss tangent has a maximum is usually defined as the glass transition temperature of the material, T_g_. As far as the dynamic property is concerned, the loss tangent denotes the damping coefficient of the material. Larger material damping is always required for controlling the vibration of structure near resonance.

Similar test results are shown in Figure 5 for Specimen T2. The monotonous decreases in E′ and a dome-shaped E″ with respect to the raise in temperature are observed, as shown in Figure 4. However, the loss tangent curve shows a plateau with two distinct peaks. Although the peak value of the loss tangent was not as high as the Specimen T1, the plateau became wider in the temperature range. In other words, Specimen T2 was more effective than Specimen T1 in reducing the vibration by material damping over a wider working temperature range. If the definition of glass transition temperature was recalled from the previous discussion, Specimen T2 had two T_g_’s at around 67 °C and 83 °C. The appearance of two loss tangent peaks is related to the HEMA toughener added in the epoxy composite, which will be discussed in more detail afterwards.

As observed from Table 2, Specimen T3 had an even higher concentration of HEMA toughener compared to the previous two specimens. The measured dynamical properties are similarly shown in Figure 6. Without too much surprise, there were also two peaks observed in the loss tangent curve with an even wider temperature range of the plateau or the lower T_g_ moving to lower temperatures. Similar measurements for the rest of the six specimens were also performed and the T_g_’s are summarized in Table 3.

### 4.2. Glass Transition Temperatures

Table 3 lists the glass transition temperatures of nine specimens obtained from the previous measurements. By cross-referencing the compositions of the specimens reported in Table 3, those listed in Table 4 with only single T_g_ are specimens without adding the HEMA toughener. The specimens with the inclusion of HEMA toughener in the compositions revealed the lower T_g_ peak. In more detail, Specimens T2, T6, and T7, which all have a 5 wt.% of HEMA, had lower T_g_’s at 67.42, 68.68, and 65.92 °C, respectively. Moreover, with further increases in HEMA content to 10 wt.%, Specimens T3, T4, and T8 lowered their corresponding T_g_’s to 52.60, 55.93, and 54.05 °C, respectively. It is known that the glass transition temperature reflects the cross-linking density of its molecular chains inside the polymer material. Those specimens without HEMA inclusion in their compositions revealed only one T_g_ peak. The introduction of a thermoplastic HEMA into the microstructure somewhat hindered the cross-linking of epoxy during curing and the molecular chains of HEMA started to move locally at lower temperature. Minor or shoulder peaks may appear due to the phase separation between the epoxy resin and the toughening agent or inhomogeneous crosslinking density [14]. However, all specimens demonstrated the T_g_ peak at higher temperature around 78.77~90.26 °C, which reflected the movement of the molecular chains in the microstructural regions unaffected by the HEMA. With more HEMA contents, the regions affected by the HEMA enlarged and, consequently, further lowered the T_g_. Of course, the movement of the molecular chain was still influenced by other nano-powders included in the composite. Therefore, the variation in T_g_ was measured for specimens with other different compositions but with less significance. The lowering of the resistance for the molecular chains to move at lower temperatures created more flexibility to the material and, therefore, alleviated the brittleness. That is the mechanism for using toughener in epoxy resin to raise its toughness. Since the main purpose of this study is to investigate the effects of the nano-powders on the dynamic property of the epoxy composite, only the second T_g_ is referred to specifically afterwards.

Table 4 lists the main glass transition temperatures of all specimens and their calculated statistical properties. According to these measurements, the S/N ratio of the glass transition temperature for each control factor was summarized and presented in Table 5. It is observed that all nano-powders had higher effect (S/N ratio) on the glass transition temperature with higher level of doping concentration. Among them, SiO_2_ had the most significant effect while Al_2_O_3_ had the least. It is worth mentioning that the amount of HEMA doping decreased the glass transition temperature drastically, which was mainly caused by the rotation and vibration of the organic molecules of HEMA toughener under the influence of heating. Moreover, the reaction from the added toughener in the epoxy also hindered its crosslinking and reduced the crosslinking density between the molecular chains. Therefore, the molecular chains had more space for moving and the T_g_ decreased accordingly. These nano-powders usually have higher stiffness than the epoxy matrix. The reinforcement demonstrates its effect with the raise in glass transition temperature. On the other hand, the HEMA toughener showed an opposite trend because of its interference with the cross-linking density, as discussed previously. Among the control factors, HEMA had the highest influence on the glass transition temperature followed by silica, alumina and carbon black, sequentially.

With the obtained experimental data presented in Table 4, a multi-variable regression analysis by using SPSS software was performed on the relationship of the glass transition temperature of the epoxy composite with respect to the four control factors. The following polynomial equation was obtained for the prediction of the glass transition temperature Y of the epoxy composite.
(8)$$Y=83.08+0.537\;X_{A}+0.183\;X_{A}^{2}+1.087\;X_{B}+0.273\;X_{B}^{2}+2.978\;X_{C}-0.832\;X_{C}^{2}-0.535\;X_{D}-0.210\;X_{D}^{2}$$

In the above equation, $X_{A}$, $X_{B}$, $X_{C}$, and $X_{D}$ are the wt.% of the silica nano-powder, alumina nao-powder, carbon black nano-powder, and HEMA toughener, respectively. In order to check the validity or accuracy for the prediction equation, the specimen with the optimal combination of the control factors (A3, B3, C3, and D1 as revealed in Table 5) to maximize the glass transition temperature was fabricated and tested. The predicted 90.61 °C from Equation (8) was close to the measured 90.91 °C. The 0.33% difference between the prediction and measurement showed good accuracy of the regression model in predicting the glass transition temperature of the epoxy composite. 

### 4.3. Storage Modulus

The storage modulus of the material used in packaging denotes the material’s measure to withstand deformation due to dynamic loading. The higher the storage modulus, the better the resistance to dynamic deformation. Table 6 presents the analysis on the measured results of the storage modulus at 30 °C for the nine specimens. Following the similar data processing of the previous glass transition temperature, the S/N ratios of storage modulus for each control factor at different levels were obtained and listed in Table 7. The results showed that the S/N ratio of the storage modulus increases with the doping content of each nano-powder. Among the three nano-powders studied, the carbon black is more effective in raising the storage modulus, while the Al_2_O_3_ has the least effect. However, the HEMA toughener played a different role of lowering the storage modulus with increasing doping content. The lowering effect was more pronounced than its nano-powder counterparts in raising the storage modulus of the epoxy composite as the S/N ratio was twice larger in magnitude. As mentioned previously, the incorporation of thermoplastic HEMA molecules into the epoxy polymer chains impedes the cross-linking and reduces the local cross-linking density. Thus, the storage modulus reduced with higher HEMA contents. On the other hand, the nano-powders filled in the epoxy matrix act as the reinforcement phase in the composite. Moreover, the dispersion of hard inorganic nano-powders in the microstructure of epoxy also serves as the pin-point sites to increase the resistance of chain movement. Thus, deformation reduced and the stiffness of the composite increased. It should also be mentioned that the insertion of HEMA molecules in blocking the crosslinking of the epoxy network and the incorporation of nano-powders in the microstructure could interfere with crack propagation in the epoxy composite. Therefore, the toughness of the epoxy composite can be enhanced.

Similar regression fit using SPSS software was conducted relative to the relationship between the storage modulus and the control factors. The storage moduli measured from the nine specimens were employed in the regression fit with the content wt.% of the control factors. The obtained equation is listed as follows.
(9)$$E^{\prime }=2066-16.667\;X_{A}+60.333\;X_{A}^{2}+130.167\;X_{B}-11.167\;X_{B}^{2}+122.500\;X_{C}+3.833\;X_{C}^{2}-17.733\;X_{D}-3.387\;X_{D}^{2}$$

In order to check the accuracy of the above prediction function for the storage modulus, a specimen with the optimal combination of control factors (A3 B3 C3 D1 from Table 6 with maximum S/N ratio) was fabricated and tested. The prediction from Equation (9) was 2749.5 MPa while the measured one was 2770.3 MPa. A small deviation of 0.76% from prediction to measurement illustrates the model depicted in Equation (9). Equation (9). can be considered as a good model in the design of this epoxy composite for use in dynamic loading environments. 

## 5. Conclusions

When material is used in vibrational environment, the capability to estimate its dynamic mechanical property is critical during the design to prevent the unexpected dynamic response from external excitation. The dynamic property such as the stiffness and damping coefficients are crucial for the packaging materials used in the delicate electronic devices. This study explored the feasibility to predict the storage modulus and glass transition temperature of an epoxy composite that incorporated the nano-powders of silica, alumina, and carbon black in a HEMA toughened epoxy matrix. The use of nano-powders as reinforcements was shown to increase both the T_g_ and the storage modulus, while the addition of HEMA toughener lowered both dynamic characteristics. With only the incorporation of each nano-powder of silica, alumina, and carbon black by 2 wt.%, the storage modulus of the epoxy composite at 30 °C showed a 34% increase from its pristine counterpart. On the other hand, the addition of 10 wt.% of HEMA into the epoxy could widen the damping plateau of the loss tangent spectrum from a temperature span around its T_g_ from 20 °C to 50 °C. This wider plateau denoted that the material damping could be operative in larger temperature span or frequency span. Finally, the prediction equations for the glass transition temperature and storage modulus were obtained from regression fits of the measured data, respectively. These equations could be used for design purpose and showed <1% deviation from the measurement within the range of control factors investigated in this study.

## Figures and Tables

**Figure 1 materials-14-04193-f001:**
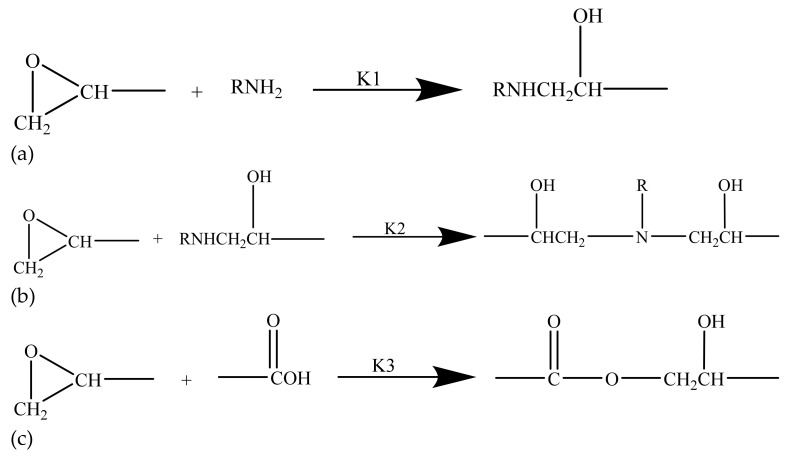
The ring-opening reactions involved in polymerization of epoxy: (**a**) the ring-opening reaction of the epoxide group and the primary amine of hardener agent; (**b**) the ring-opening reaction of the epoxide group with the secondary amine formed in (**a**); (**c**) the self-catalyzed ring-opening reaction from the hydroxyl group.

**Figure 2 materials-14-04193-f002:**
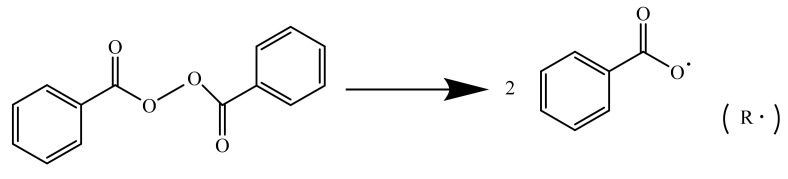
The disintegration reaction of benzoyl peroxide into free radical.

**Figure 3 materials-14-04193-f003:**
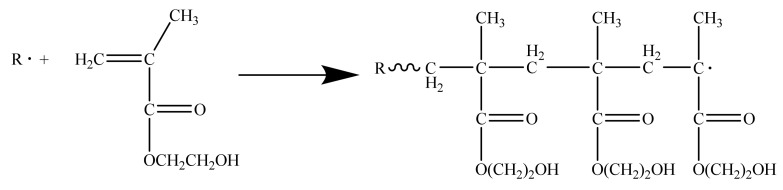
The polymerization reaction of 2-hydroxyethyl methacrylate to form the PHEMA polymer.

**Figure 4 materials-14-04193-f004:**
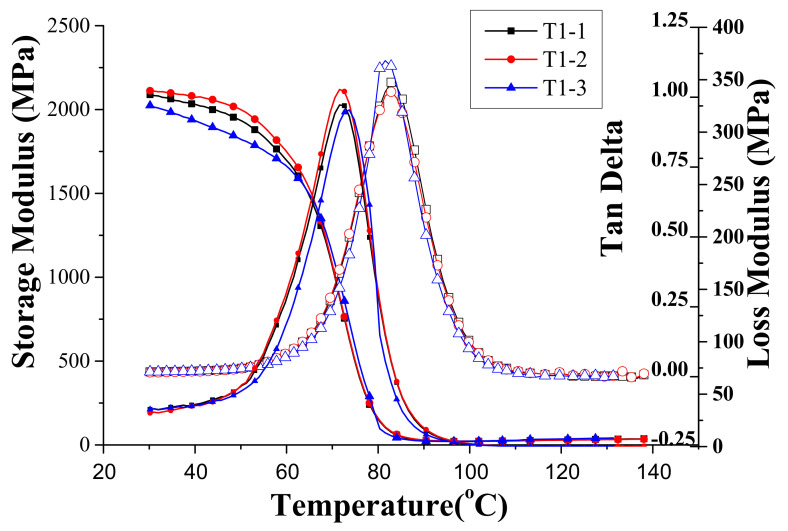
The dynamic properties of Specimen T1 measured with the temperature sweeping from 30 °C to 140 °C.

**Figure 5 materials-14-04193-f005:**
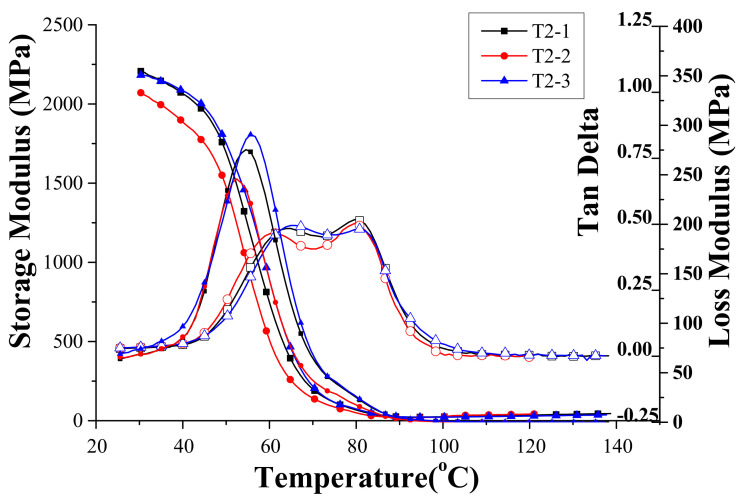
The dynamic properties of Specimen T2 measured with the temperature sweeping from 30 °C to 140 °C.

**Figure 6 materials-14-04193-f006:**
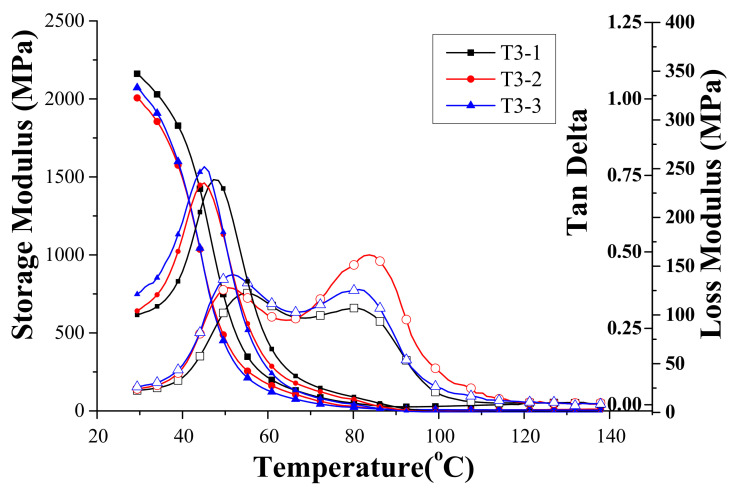
The dynamic properties of Specimen T3 measured with the temperature sweeping from 30 °C to 140 °C.

**Table 1 materials-14-04193-t001:** L_9_(3^4^) orthogonal table for the design of experiments.

No. of Specimen	Factor A	Factor B	Factor C	Factor D
1	1	1	1	1
2	1	2	2	2
3	1	3	3	3
4	2	1	2	3
5	2	2	3	1
6	2	3	1	2
7	3	1	3	2
8	3	2	1	3
9	3	3	2	1

**Table 2 materials-14-04193-t002:** The compositions of the specimens used in an L9 orthogonal array.

Specimen	Epoxy	Hardener	Al_2_O_3_	SiO_2_	Carbon Black	HEMA	BPO
	(wt.%)	(wt.%)	(wt.%)	(wt.%)	(wt.%)	(wt.%)	(wt.%)
T1	74.64	24.86	0	0	0	0	0
T2	69.04	23.02	0	1	1	5	0.5
T3	63.46	21.19	0	2	2	10	0.5
T4	65.27	21.74	1	0	1	10	0.5
T5	70.97	23.71	1	1	2	0	0
T6	68.18	22.68	1	2	0	5	0.5
T7	67.23	22.42	2	0	2	5	0.5
T8	64.37	21.44	2	1	0	10	0.5
T9	70.14	23.35	2	2	1	0	0

**Table 3 materials-14-04193-t003:** Glass transition temperatures of the specimens.

Specimen	T_g_1 (°C)	T_g_2 (°C)
T1	-	83.08 ± 0.31
T2	67.42 ± 1.13	83.34 ± 0.80
T3	52.60 ± 2.28	81.34 ± 1.44
T4	55.93 ± 3.29	78.43 ± 1.76
T5	-	87.63 ± 1.13
T6	68.68 ± 2.29	83.86 ± 0.47
T7	65.92 ± 2.29	84.15 ± 0.42
T8	54.05 ± 1.69	78.77 ± 1.19
T9	-	90.26 ± 2.50

**Table 4 materials-14-04193-t004:** Analysis on the measured results of the main glass transition temperature.

Specimen	Y_1_ (°C)	Y_2_ (°C)	Y_3_ (°C)	Y_Avg_ (°C)	C.O.V. (%)	S/N (dB)
T1	82.84	82.94	83.45	83.08	0.39	38.39
T2	82.31	83.91	83.81	83.34	1.08	38.42
T3	80.14	83.01	80.87	81.34	1.83	38.20
T4	76.44	78.91	79.95	78.43	2.30	37.89
T5	86.71	88.96	87.21	87.63	1.35	38.85
T6	84.46	83.52	83.59	83.86	0.62	38.47
T7	83.88	83.87	84.71	84.15	0.57	38.50
T8	77.21	79.52	79.58	78.77	1.72	37.92
T9	91.54	87.12	92.11	90.26	3.03	39.10

**Table 5 materials-14-04193-t005:** Reaction table for the S/N ratio of glass transition temperature (unit: dB).

Control Factor	Al_2_O_3_	SiO_2_	Carbon Black	HEMA
Level 1	38.34	38.26	38.26	38.78
Level 2	38.40	38.40	38.47	38.46
Level 3	38.51	38.59	38.52	38.00
Effect	0.17	0.33	0.26	0.78

**Table 6 materials-14-04193-t006:** Analysis on the measured results of the storage modulus at 30 °C.

Specimen	E′_1_ (MPa)	E′_2_ (MPa)	E′_3_ (MPa)	E′_Avg_ (MPa)	C.O.V. (%)	S/N (dB)
T1	2081	2107	2009	2066	3.21	66.30
T2	2184	2051	2178	2138	3.12	66.59
T3	2094	1969	2016	2026	3.26	66.13
T4	1688	1709	1764	1720	3.76	64.71
T5	2516	2485	2465	2489	2.73	67.92
T6	2166	2118	2173	2152	3.10	66.66
T7	2373	2285	2425	2361	2.86	67.45
T8	1963	1849	1818	1877	3.49	65.45
T9	2618	2557	2672	2616	2.61	68.35

**Table 7 materials-14-04193-t007:** Reaction table for the S/N ratio of storage modulus (unit: dB).

Control Factor	Al_2_O_3_	SiO_2_	Carbon Black	HEMA
Level 1	66.34	66.15	66.14	67.52
Level 2	66.43	66.65	66.55	66.90
Level 3	67.09	67.04	67.17	65.43
Effect	0.75	0.89	1.03	2.09

## Data Availability

The data underlying this article will be shared on reasonable request from the corresponding author.

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
