# Peer review of "Improvement Prediction on the Dynamic Performance of Epoxy Composite Used in Packaging by Using Nano-Particle Reinforcements in Addition to 2-Hydroxyethyl Methacrylate Toughener"

_materials, 2021, doi:10.3390/ma14154193_

Round 1

Reviewer 1 Report

The paper deals with the analysis of compositions' effects of nanoparticles on some epoxy resins nanocomposites. Even if experiments were sensibly arranged and well described in the manuscript no new properties have been reported. Furthermore, no data on the dispersion degree of the nanoparticles has been reported. On the contrary, this parameter is very important to evaluate the effectiveness of the loading on the observed composite properties. 

Conclusions are well supported by data. However, the novelty is below the expected merit for publication in Materials. 

Author Response

Ans: Thank you very much for the comments! Regarding the dispersion of the added nano-powders in the epoxy, it has been believed that the ultrasonic and centrifugal mixing is able to help the well dispersion of these nano-powders in our previous study [18]. The citation of this work has been added in the revised manuscript accordingly. Moreover, the novelty of this manuscript lies on the report of an experimentally based formula for the prediction of the dynamic mechanical property of a nano-powder reinforced epoxy for design applications. It is believed that the result can be helpful to the material engineers in the electronic packaging industry.

Reviewer 2 Report

This manuscript investigated the influence of degree and composition of nine kinds of epoxy composites on dynamic mechanical parameters. The author found that, there is possibility of anticipation the storage modulus and glass transition temperature of an epoxy composite which incorporated nano-powders such as silica, alumina and carbon black in a HEMA toughened epoxy matrix. The topic of this research was important, and the results were also interesting; however, the quality of this manuscript should be further improved before acceptance. I have the following suggestions for improvement of the manuscript. It should be noted that the level of English is very high, which contributes to a positive reception of manuscript.

  1. Line 309-310: “According to these measurements, the S/N ratio of the glass 309 transition temperature for each control factor was summarized and presented in Table 5” the S/N ratio and other parameters scholud be explained.
  2. The results presented in Tables 5 and 7 should be discussed in the paragraph.
  3. Descriptions in Table 6 are unclear, in my opinion the Y should be convert to E’.
  4. The quality of chemical formulas in manuscript should be corrected. Chemical formulas contain atoms of different sizes, the angles between the atoms incorrect.

Author Response

  1. Line 309-310: “According to these measurements, the S/N ratio of the glass 309 transition temperature for each control factor was summarized and presented in Table 5” the S/N ratio and other parameters scholud be explained.

Ans: Thank you very much for the comments! More explanations about the possible mechanisms for the different influence of the nano-powders and toughener have been added in the revised manuscript as suggested.

      2.The results presented in Tables 5 and 7 should be discussed in the paragraph.

Ans: Thank you very much for the comments! More discussion on the opposite effects of HEMA toughener and nano-powders on the storage modulus has been assed in the revised manuscript as suggested.

     3.Descriptions in Table 6 are unclear, in my opinion the Y should be convert to E’.

Ans: Thank you very much for the suggestion! The notations for the storage moduli of three tested specimens have been changed to E’ in Table 6.

     4.The quality of chemical formulas in manuscript should be corrected. Chemical formulas contain atoms of different sizes, the angles between the atoms incorrect.

Ans: Thank you very much for the comment! The expressions in Eqns. (1) ~ (5) have been corrected in the revised manuscript.

Reviewer 3 Report

In the paper from Chen et al. entitled "Improvement of prediction on the dynamic performance of epoxy composite used in packaging by using nano-particle reinforcements in addition to 2-hydroxyethyl methacrylate toughener" the authors use nanoparticles of alumina, silica and carbon black, together with as toughening agents for epoxy composites. The authors provide sufficient information about the nanoparticulate materials to enable reproduction of the results. Upon minor English revision, the paper can be accepted as it is.

Author Response

Ans: Thank you very much for the recognition and comment! The English writing of the manuscript has been proofread carefully again.

Round 2

Reviewer 1 Report

The performed revision improved the quality of the manuscript that now is considered acceptable for publication